# Microwave Spoof Surface Plasmon Polariton-Based Sensor for Ultrasensitive Detection of Liquid Analyte Dielectric Constant

**DOI:** 10.3390/s21165477

**Published:** 2021-08-13

**Authors:** Ivana Podunavac, Vasa Radonic, Vesna Bengin, Nikolina Jankovic

**Affiliations:** BioSense Institute, University of Novi Sad, Dr Zorana Djindjica 1, 21000 Novi Sad, Serbia; vasarad@biosense.rs (V.R.); bengin@biosense.rs (V.B.); nikolina@biosense.rs (N.J.)

**Keywords:** microwave sensor, spoof surface plasmon polariton (SSPP), permittivity sensing, edible oils

## Abstract

In this paper, a microwave microfluidic sensor based on spoof surface plasmon polaritons (SSPPs) was proposed for ultrasensitive detection of dielectric constant. A novel unit cell for the SSPP structure is proposed and its behaviour and sensing potential analysed in detail. Based on the proposed cell, the SSPP microwave structure with a microfluidic reservoir is designed as a multilayer configuration to serve as a sensing platform for liquid analytes. The sensor is realized using a combination of rapid, cost-effective technologies of xurography, laser micromachining, and cold lamination bonding, and its potential is validated in the experiments with edible oil samples. The results demonstrate high sensitivity (850 MHz/epsilon unit) and excellent linearity (R^2^ = 0.9802) of the sensor, which, together with its low-cost and simple fabrication, make the proposed sensor an excellent candidate for the detection of small changes in the dielectric constant of edible oils and other liquid analytes.

## 1. Introduction

Due to their possibility for real-time, non-contact and non-invasive measurements, microwave sensors present an excellent solution for a wide range of applications, including dielectric constant sensing [1,2,3,4,5,6,7,8,9], food quality control [10,11,12,13,14,15,16,17,18,19], gas sensing [20,21,22], detection of biomolecules [23,24], glucose monitoring [25,26], measurements of concentration for liquid solutions [27,28], microwave imaging [29,30], and mechanical motion sensing [31,32]. They can also be combined with other technologies, including microfluidics, which provide compact and cost-effective platforms for rapid detection in small amounts of liquid samples [33,34,35,36]. Although there have been a number of attempts to increase their sensitivity, including employment of split-ring resonators (SRR) [3,14,19,37], transmission lines [15], meta-surface absorbers [16], patch resonators [18], most microwave sensors do not have the ability to detect small changes in dielectric constant, and they are usually proposed for detection in the wide range of values, for applications that do not require accurate detection.

Another promising phenomenon for sensing solutions are surface plasmon polaritons (SPPs), surface waves that naturally occur at the conductor/dielectric interface at optical frequencies. Many unique optical properties of SPPs, such as localized field enhancement and light confinement at the subwavelength scale enabled the realization of miniaturized and highly sensitive sensors suitable for different applications [9,38,39,40,41,42,43]. However, SPPs are not supported at far-infrared, microwave, and terahertz frequencies, and the concept of spoof surface plasmon polaritons (SSPPs) was introduced by using designed structures and plasmonic metamaterials [44,45] with the aim to mimic SPP propagation in those frequency ranges. Different planar designs that support SSPP propagation were proposed in the literature, based on gradient holes [46], corrugations [47,48], zigzag grooves [48], grooves with circular patches [49], and disk resonators [50,51]. In addition, SSPP properties were used for the realization of filters [52,53,54], antennas [55,56] and waveguide systems [57,58,59]. In spite of their high potential for sensing applications, only a few sensors based on SSPP were proposed in the literature including refractive index and thickness sensor [60], a sensor for human skin tissue water content [61], and the detector of Schottky diode [62]. All of the proposed SSPP based sensing solutions [60,61,62] use comb structure unit cells with highly sensitive properties. The sensor for human skin tissue water content [61] based on planar plasmonic waveguide was realized in printed circuit board (PCB) technology and showed the ability for in vivo measurements of the water content in the skin tissue with potential for early diagnostic applications. On the other hand, the Schottky diode detector based on SSPP was also realized in PCB technology where the SSPP configuration enabled a significant increase in detection sensitivity [62]. Although the sensing solution based on metamaterial corrugated metal stripe structure showed ultrasensitive properties for thickness and refractive index sensing, the sensor was not realized experimentally [60]. In addition, SPP-like phenomena were proposed for dielectric constant sensing and the sensor with highly sensitive performances was realized in printed circuit board (PCB) technology [9].

One of the potential applications of microwave sensors in which the sensitive response can provide valuable information about the sample is edible oil quality control. In that sense, dielectric spectroscopy has been used for the detection of adulteration [63], chemical composition [64,65], quality of edible oils [13], frying oil degradation [11,12], characterization of cooking oils [14], usage time of oil [15], and identification of oils [16,17], however, usually lacking high sensitivity and accuracy.

In this paper, we bring together the advantages of microwave and SSPP phenomena as well as the microfluidics concept and realize a very sensitive and low-cost sensor for detection of small changes in the dielectric constant of liquids. First, a novel compact unit cell was proposed, which enables SSPP propagation and strong field confinement in the microwave regime. Based on that, a multi-layer structure that combines the proposed SSPP microwave configuration with a microfluidic channel is proposed, consequently providing a sensing platform for liquids. The proposed sensor is realized using low-cost materials and manufacturing technologies of xurography and laser micromachining, which has been rarely employed in sensor design despite its advantages. To validate the sensing potential in a real-world problem, dielectric constants of edible oil samples are detected and compared to those from the literature. It is shown that the sensor exhibits excellent performance, in particular in terms of fabrication complexity, cost, and sensitivity.

The paper is organised into five sections. After the introduction with a detailed literature overview and the topic description, Section 2 describes the sensor operating principle, its modelling, and simulations. Realization of the sensor through materials and method description is presented in Section 3. The experimental results and comparison with simulations are presented in Section 4, as well as discussion and comparison of the proposed sensor with other sensing solutions proposed for similar applications. Finally, Section 5 presents a summarized conclusion for this research and future research directions.

## 2. Theoretical and Numerical Analysis

The layout of the proposed sensor is shown in Figure 1. Its operating principle is based on the SSPP structure, which is realized using conductive layers 1 and 7, whereas layers 2–6 serve as dielectric substrates and simultaneously host a microfluidic reservoir. Due to the fabrication procedure, which will be explained in detail in the following section, the layers are made of aluminium (Al) (layers 1 and 7), Polyvinyl Chloride—PVC (layers 2 and 6), Poly (methyl methacrylate)—PMMA (layer 4), and 3 M double-sided adhesive tapes (layers 3 and 5). Moreover, conductive layers have been optimized to have a smooth transition between microstrip lines and ports.

The core element of the sensor is the SSPP structure, i.e., a unit cell, of which the geometrical properties dictate the behaviour of the SSPP structure. Namely, the SSPP phenomenon in such circuits is provided owing to the array of cells, which enables slow-wave behaviour of the structure at the frequencies that correspond to the resonant behaviour of the cell. In other words, at the frequencies that are close to the resonant frequency of unit cell, the wave becomes more confined, and the corresponding dispersion characteristics start to deviate from the light line, finally to reach the frequency that does not allow propagation, and which is characterized by a strong transmission zero in the response of the SSPP structure.

In that sense, first a novel unit cell is proposed, Figure 2a, which provides a more compact structure in comparison to the conventional comb-like one, Figure 2b, owing to the two symmetric bent structures that form a T-shaped gap. If we consider the unit cells as resonant cells with their effective inductance and capacitance, one can note that while the proposed cell has a similar effective capacitance towards the ground layer as the comb-like one, its effective inductance is significantly higher, and consequently, the proposed cell exhibits lower resonant frequency.

To validate the previous statement, dispersion diagrams of the two cells with the same footprint area were examined. The comb-like unit cell is characterized with four parameters presented in Figure 2b—period of unit cell *d*, the distance between grooves *a*, depth of groove *h*, the width of the structure *w*, and their relations *a* = 0.4*d*, *h* = 0.8*d*, *w* = *d*, while the proposed unit cell is characterised by two additional geometrical parameters—width of the bent structures *b* = 0.67*d*, and distance between bent shapes and groove depth *x* = 0.2(*w − h*). It can be noticed that all proposed parameters are a function of unit cell period *d* which enables scaling the structure. The dispersion diagrams were obtained in CST Microwave Studio 2021^®^
*Eigenmode* solver, using corresponding materials for conductive and dielectric layers as explained at the beginning of this section. The conductive layers are modelled as 40-µm-thick Al with electrical conductivity of 3.56 × 10^7^ S/m. Dielectric constant 3.3 and tanδ 0.15 were used for modelling 80-µm-thick PVC foil and, finally, 2-mm-thick PMMA was modelled with dielectric constant 2.6 and tanδ 0.02. Although it makes a negligible contribution to the results, the 3M double-sided adhesive tapes were modelled as thin films with dielectric constant equal to 3. One should note that the microfluidic reservoir, i.e., the liquid sample in the final sensor, which is a part of the dielectric substrate of the unit cell, is initially set to have the dielectric constant $\varepsilon_{r}$ = 1.

Figure 2c presents the comparison of the dispersion diagrams for different values of unit cell period *d*. Simulation results show that dispersion diagrams for both unit cells deviate significantly from the light line, which indicates slow-wave behaviour and strong field confinement. In both cases, an increase of the period *d* leads to a decrease in surface plasmon frequency, however, the proposed cell exhibits lower surface plasmon frequencies in all cases of *d*, which is due to the higher effective inductance and consequently lower resonant frequency.

Since the sensor is based on the SSPP structure, a numerical analysis is done to investigate the sensitivity of the surface plasmon frequency of the two-unit cells, to the dielectric constant of the liquid sample in the microfluidic reservoir. For the sake of simplicity, the dielectric constant is varied from 1 to 5, which approximately corresponds to the values of dielectric constants of edible oils, i.e., real samples that are used in the experiments in the following section. The samples are modelled without considering losses due to the generality of analysis and optimisation of sensing performances. Also, the unit cell period of *d* = 7.5 mm is chosen as an optimal value, which can provide compact dimensions and simple realization in the proposed hybrid fabrication technology.

The results in Figure 2d present a comparison of dispersion diagrams for the two-unit cells for different values of the dielectric constant of the sample. Additionally, the summary graph for the surface plasmon frequency dependence of the sample’s dielectric constant is given in Figure 2e. As the dielectric constant is increased, surface plasmon frequency shifts to the lower frequencies. The proposed unit cell has better linearity (R^2^ = 0.9756) than the comb-like unit cell, as well as greater compactness. In addition, for the range of dielectric constants from 1 to 2 where the values of edible oils in the microwave range are expected, the sensitivity of the sensor based on the proposed unit cell was estimated to 850 MHz/epsilon unit.

Although one may argue that the comb-like unit cell has better sensitivity, i.e., higher frequency shift for the same range of the dielectric constants, it should be noted that the proposed unit cell exhibits significantly better linearity, which is another important property for the sensing applications. Therefore, the proposed cell exhibits better overall sensing potential than the comb-like one.

To confirm the behaviour and potential of the proposed unit cell, corresponding SSPP sensor structure has been simulated in CST software. As indicated in Figure 1, the SSPP structure has been optimized and consists of seven-unit cells that are connected to the microstrip line, where impedance matching with the input and output ports have been achieved using tapers. The dielectric substrate consists of five layers so it can host a microfluidic reservoir, which is positioned below the array of the unit cells. The final dimensions of the overall structure are as follows: the whole sensor dimensions 120 mm × 22.5 mm, a microfluidic reservoir 52.5 mm × 7.5 mm, and the width of microstrip line widens from 1 mm to 5 mm where the first corresponds to the input of surface mount assembly (SMA) connectors.

Figure 2f shows responses of the SSPP sensor structure for the different dielectric constant of the liquid sample in the microfluidic reservoir, and one can note narrow and deep transmission zeros that correspond to the surface plasmon frequencies. Namely, at the surface plasmon frequency, the group velocity tends to zero, so there is no wave propagation, which creates a transmission zero in the response. A small disagreement of the exact position of the transmission zeros and surface plasmon frequencies is due to the finite dimensions of the SSPP structures, contrary to infinite arrays of unit cells that were used for numerical simulations of dispersion diagrams.

It should be noted that the presented sensor dimensions have been optimized for excellent sensing properties for the narrow range of dielectric constants of different oil samples. Nevertheless, the sensor can be adjusted to different and wider ranges of dielectric constants. To that end, the configuration should be modified and finely tuned in terms of unit cell dimensions and substrate parameters, which also opens a possibility for further sensor miniaturization. The previous discussion reveals a strong potential of the proposed unit cell and corresponding SSPP structure for sensitive, accurate, and linear sensing of small changes in dielectric constant, whose operating principle is based on the transmission zero shift in the SSPP sensor response. The following section will experimentally confirm this statement, and also show that the proposed sensor is characterised by a low-cost and simple fabrication procedure.

## 3. Materials and Methods

The proposed sensor consists of a multilayer structure, and thus it requires a multistep fabrication process. Nevertheless, the proposed fabrication procedure is based on the technologies of xurography and laser micromachining with low-cost materials and thus can be considered as simple, cost-effective, and quick since the overall procedure lasts only several minutes.

Conductive layers 1 and 7 are made of aluminium sticky foils and cut with Nd:YAG laser Rofin-Sinar Power Line D-100 laser. Layer 4 with a microfluidic reservoir is made of PMMA, by using CO_2_ laser CNC—MBL 4040RS. Layers 2 and 6 are made as PVC foils (MBL 80MIC Belgrade, Serbia), while layers 3 and 5 are 3M double-sided adhesive tapes, and they are all cut with Plotter Cutter Roland DG CAMM-1 GS-24. In the cold lamination process, all layers were bonded in the order presented in Figure 1.

The layout of each fabricated layer is shown in Figure 3. The same design was used for the fabrication of PMMA and 3M tape layers, presented in Figure 3a,b, respectively. Top PVC foil, presented in Figure 3c, contains inlet/outlet holes for filling the microfluidic reservoir with samples, while bottom PVC foil (Figure 3d) is used for closing the channel system. Finally, the realized sensor structure is presented in Figure 3e,f in the top and bottom view, respectively.

As stated previously, the aim of the proposed sensor is to be applied for a real-world problem, and thus it is validated through the measurement of the dielectric constants of edible oils, which represents a technique to detect oil quality and adulteration. To that end, the following samples have been used: palm oil (Palm oil for frying, Dijamant, Zrenjanin, Serbia), sunflower oil (Edible Refined oil, Bas Bas, Zrenjanin, Serbia), castor oil (Livsane, Novi Sad, Serbia), and olive oil (Cadel Monte, 100% Italian Extra Virgin Olive Oil). Besides oil samples available on the market, additional 4 samples are prepared by mixing palm and castor oils, and their dielectric constants are calculated according to the Kraszewski formula [66]:(1)$$\varepsilon_{rMIX}={\left(\sqrt{\varepsilon_{rp}}\upsilon_{p}+\sqrt{\varepsilon_{rc}}\upsilon_{c}\right)}^{2}$$
here $\varepsilon_{rMIX}$ presents dielectric constant of oil mixture, $\varepsilon_{rp}$ and $\varepsilon_{rc}$ present dielectric constant of palm and castor oil, respectively, and $\upsilon_{p}$ and $\upsilon_{c}$ fractional volume of palm and castor oil in the mixed sample. Values of dielectric constants for used oils and calculated values for samples with mixed palm and castor oils are listed in Table 1.

The SSPP sensor was filled with the oil samples starting with palm oil, which has the lowest value of the dielectric constant of the prepared samples. In each following step, the reservoir was rinsed using the sample with the first higher dielectric constant and then filled with the same sample. It should be noted that the microfluidic reservoir has a small volume and thus only 0.8 mL of the sample volume is needed. The responses of the sensor have been measured using a vector network analyser (VNA) E5071C Agilent Technology and surface mount assembly (SMA) connectors (SMA Southwest Microwave 292-04A-5) were used for connection between VNA and SSPP sensor, Figure 4. One-point calibration of the sensor was done with the microfluidic reservoir filled with air.

## 4. Results and Discussion

Figure 5 and Figure 6 present simulated and measured responses for pure edible oil samples and mixed oil samples, respectively. One should note that the simulated responses have been obtained using the calculated dielectric constants in Table 1. It can be seen that spectral positions of transmission zeros are shifted towards lower frequencies for increasing values of samples’ dielectric constants. In addition, a good agreement between simulation and measurement results can be observed, except for a small spectral shift and higher losses, where the latter can be attributed to the losses in cables and connectors, and imperfection of fabrication procedure. Since the spectral shift is small, it does not influence the sensor performances to a great extent. 

The measured results confirm that the proposed sensor is very sensitive to small changes in the dielectric constant of the sample. This is further confirmed in Figure 7 that shows the dependence of the spectral position of the simulated and measured transmission zeros to the dielectric constant of the samples. Both simulated and measured results have excellent linear properties R^2^ = 0.9971 and R^2^ = 0.9802, respectively. The sensitivity of the sensor is calculated as $\frac{f_{max}-f_{min}}{\varepsilon_{max}-\varepsilon_{min}}$ where $f_{max}$ and $f_{min}$ correspond to the frequencies of transmission zeros for samples with highest and lowest dielectric constant $\varepsilon_{max}$ and $\varepsilon_{min}$, respectively, and it is equal to 850 MHz/epsilon unit.

Therefore, the proposed sensor has a potential for application in oil quality control since it is able to distinguish oils with a difference in dielectric constant equal to 0.02. Moreover, it exhibits excellent linearity and accuracy, as well as cost-effectiveness. 

To further illustrate the excellent sensing performances of the SSPP sensor, the comparison to other sensors for dielectric constant sensing in edible oils is shown in Table 2. 

In terms of complexity, it should be noted that the hybrid approach for sensor fabrication enabled its simple and rapid preparation that can be done in a few minutes. Moreover, the proposed sensor uses less than 1 mL of sample, which reduces the amount of used samples comparing to sensors that have to be immersed during testing. Additionally, the sensor provides high sensitivity, and while the structures in [10,14] have better sensitivity, they require more complex fabrication procedures and operations at higher frequencies, consequently requiring more expensive equipment. Furthermore, the structure in [14] has not been confirmed experimentally. In addition, the proposed sensor has an excellent linearity, which outperforms linearities of other solutions.

Therefore, the proposed sensor presents the first SSPP-based sensing solution for edible oil sensing, and it provides a very sensitive, accurate, and linear response. Together with a quick and simple fabrication procedure, these characteristics make the proposed sensor an excellent candidate for sensing small changes of dielectric constant, not only of edible oils, but also for other liquid analytes.

## 5. Conclusions

In this paper, a very sensitive and low-cost SSPP-based sensor was proposed for the detection of small changes in the dielectric constant of liquids. It is based on a novel compact unit cell, whose behaviour and sensing potential were analysed in detail. The unit cell is a core element of the SSPP microwave configuration which was combined with a microfluidic channel in a multilayer structure, providing a sensing platform for liquids. The sensor was realized in the hybrid fabrication technology that combines xurography and laser micromachining processes and lamination of the realized layers. The proposed technology enables the rapid and simple realization of complex structures and provides reliable and reproducible structures. However, like in every top-down fabrication approach, the size limitations of the technology are determined with the tool used for cutting, i.e., the laser beam width, which limits the size of the realized structures. In addition, further sensor integration into complex systems and devices may require the change of fabrication technology. To validate the sensing potential in a real-world problem, dielectric constants of edible oil samples were detected and compared to those from the literature. It was shown that the sensor exhibits excellent performance outperforming other solutions in terms of fabrication complexity, cost, and sensitivity. In addition, it is shown that sensing based on the SSPP principle opens a possibility for very sensitive applications, and by changing unit cell dimensions or substrate, the proposed sensor can be also adapted for precise measurements of gas or liquid mixtures, control of petroleum product, and food quality control. Therefore, future research will be focused on the novel sensing applications of the SSPP phenomena, as well as the sensor integration into multifunctional systems.

## Figures and Tables

**Figure 1 sensors-21-05477-f001:**
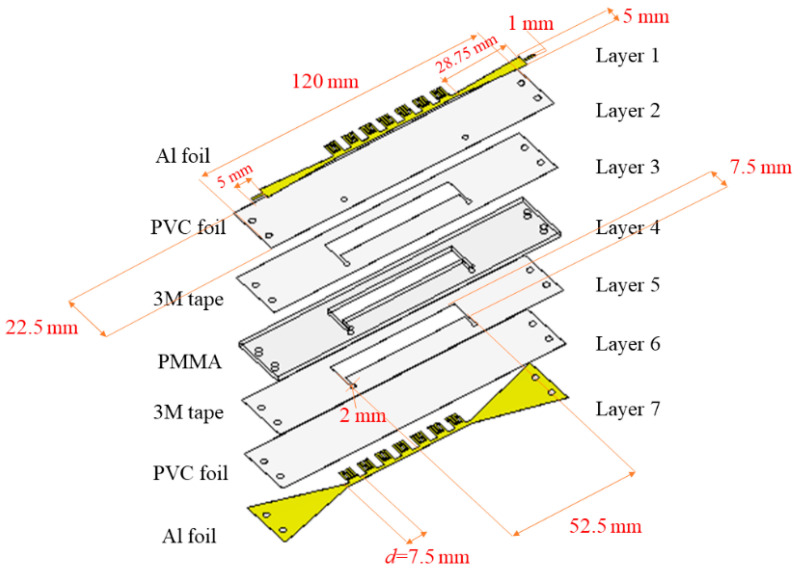
Multilayer structure of the proposed SSPP microwave sensor for dielectric constant sensing.

**Figure 2 sensors-21-05477-f002:**
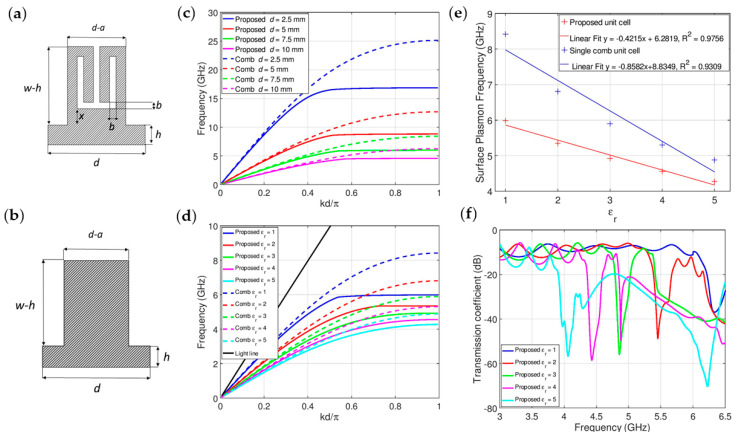
(**a**) Proposed unit cell; (**b**) unit cell of comb-like structure; (**c**) dispersion curves of comb-like and proposed unit cells for values of period *d* in the range 2.5–10 mm; (**d**) dispersion curves of comb-like and proposed unit cells with period *d* = 7.5 mm for different values of dielectric constant of sample *ε_r_* = 1–5; (**e**) surface plasmon frequencies dependence on dielectric constant for the proposed and comb-like unit cells. (**f**) Spectral positions of transmission zeros for final sensor structure.

**Figure 3 sensors-21-05477-f003:**
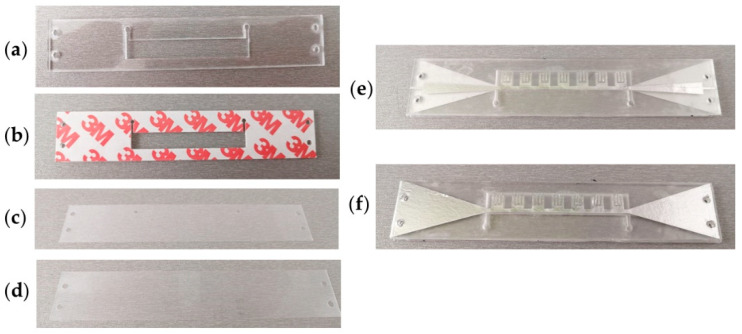
Fabricated layers of SSPP sensor; (**a**) PMMA layer with the microfluidic reservoir; (**b**) 3M double-sided adhesive tape; (**c**) top layer made in PVC foil; (**d**) bottom layer made in PVC foil; (**e**) Layout of the final structure—top view; (**f**) layout of the final structure—bottom view.

**Figure 4 sensors-21-05477-f004:**
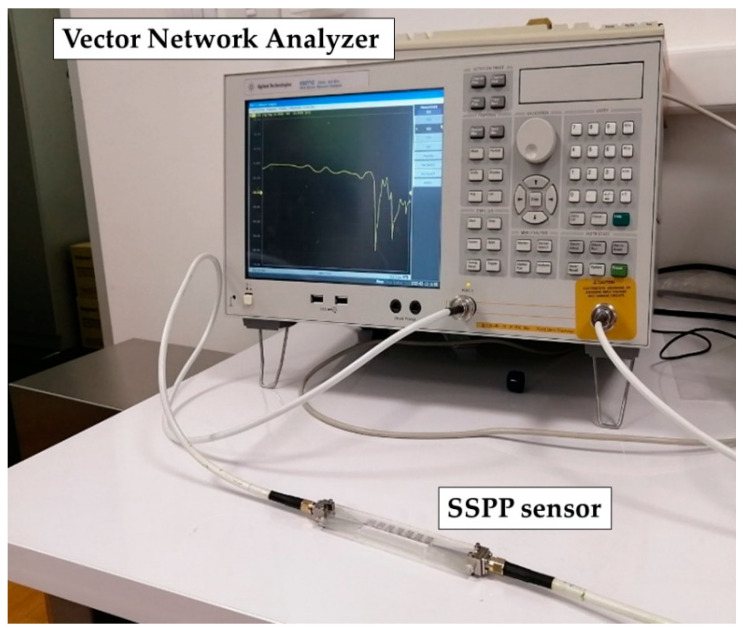
Measurement setup for the proposed sensor.

**Figure 5 sensors-21-05477-f005:**
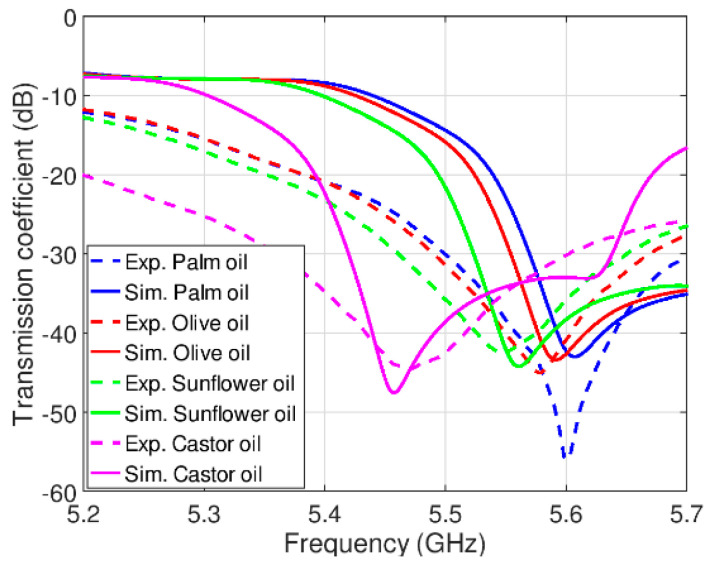
Simulation and experimental results for edible oil samples.

**Figure 6 sensors-21-05477-f006:**
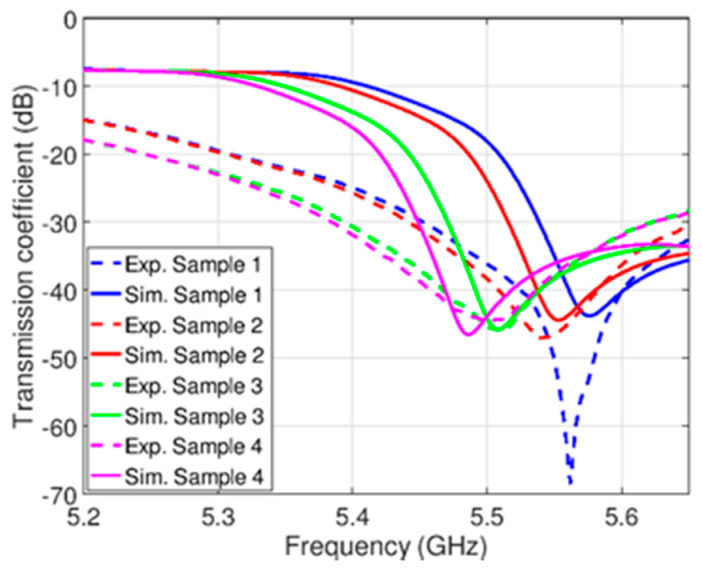
Simulation and experimental results for mixed palm and castor oil samples as listed in Table 1.

**Figure 7 sensors-21-05477-f007:**
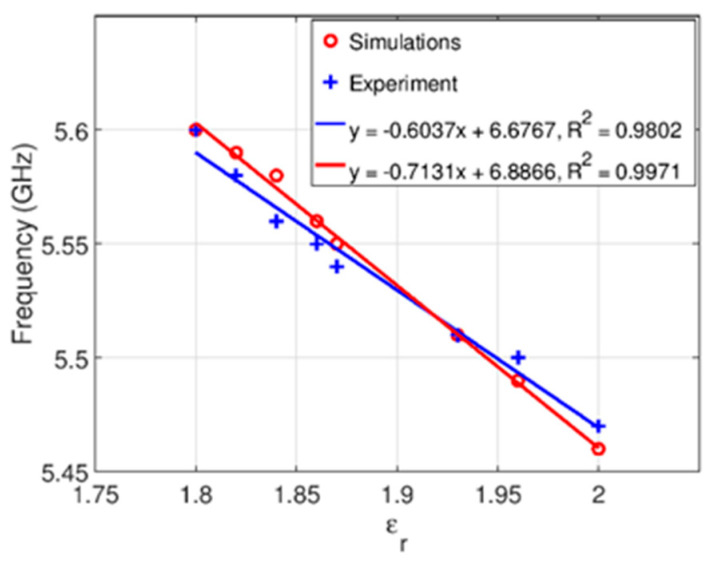
Simulated and experimental spectral positions of transmission zeros for all measured samples.

**Table 1 sensors-21-05477-t001:** Dielectric constants from literature and calculated values for oil mixtures.

Sample Number	Diel. Const.	tanδ	Reference
Palm Oil	1.80	0.03	[67]
Olive oil	1.82	0.04	[68]
Sunflower oil	1.86	0.03	[69]
Castor oil	2	0.08	[70]
Mixed samples	Diel. Const.	tanδ	Ratio-Castor: Palm oil
Sample 1	1.84	0.04	1:3.5
Sample 2	1.87	0.04	1:2
Sample 3	1.93	0.06	2:1
Sample 4	1.96	0.07	3.5:1

**Table 2 sensors-21-05477-t002:** Comparison between recently proposed sensors for edible oil sensing with the proposed SSPP sensor.

Application	Design	Technology	Technology Complexity	Sample Volume	Sensitivity (MHz/$\varepsilon_{r}$)	Operating Frequency	Ref.
Adulteration detection	Double complementary SRR	PCB	Medium	Immersed	1133	11.56 GHz	[10]
Characterization of Cooking oils	Metamaterial—SRR	Designed	Medium	Estimated: 0.01 mL	1120	30 GHz	[14]
Determination of frying time	Transmission line—sensor 1	PCB	Medium	Immersed	Estimated 243	Estimated 5.45 GHz	[15]
Determination of frying time	Transmission line—sensor 2	PCB	Medium	Immersed	Estimated 270	Estimated 5.45 GHz	[15]
Identification of edible oils	Meta-surface absorber	PCB	Medium	Estimated <1 mL	500	9.887 GHz	[16]
Adulteration detection	EBG-inspired Patch resonator	PDMS	High	0.8 mL	205.1	2.592 GHz	[18]
Dielectric characterization	Double SRR	PCB	Medium	Immersed	74.37	1.85 GHz	[19]
Proposed sensor	SSPP	Hybrid	Simple	0.8 mL	850	6.32 GHz	-

## Data Availability

The data are contained within the article.

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
