# Peer review of "Microwave Spoof Surface Plasmon Polariton-Based Sensor for Ultrasensitive Detection of Liquid Analyte Dielectric Constant"

_sensors, 2021, doi:10.3390/s21165477_

Round 1

Reviewer 1 Report

This paper designed a SSPP based biosensor for the oil detection. It is very interesting for the microwave, millimeter wave and terahertz bio-sensing. However some comments should be considered for further improvement.

1, The title shows the bio-sensor is used for the liquid analyte detection, however in the text, only four oil samples is measured.

2, Section 2, Fig.1, it is a principle prototype. I want to know how to integrate it into the final devices or equipment?

3, Fig.2, for the proposed biosensor structure, it seems the advantage of compactness is not very obvious, which is given in line 95 page 2. For the proposed structure, such as d=2.5mm, its bandwidth is much wider than the comb one, but from Fig.2(f), the authors just use a narrow band (3-5.6GHz). So I don't know the proposed structure is compact or not.

4, Page 6, the sensor is measured with oil samples. The dielectric constant is only 1.8-2.0. So I want to know how is the performance if the liquid is changed to some other samples such as water?

5, Table 1, the dielectric constant and loss is estimated. Please give the data support. 

6, Fig.6, the simulation and measured results agree not well. Please explain the reasons.

7, Fig.7, the results are fitted from the transmission zeros with samples of oil. If the samples is changed to some other liquid (its dielectric constant may change) or the oil mixture is in-homogeneous, the transmission may disappear. So the bio-sensor will not work well.

8, Ref.[29], the format needs to be improved.

Author Response

Dear Sir/Madam,

We appreciate the careful reviewing of our manuscript and we gratefully acknowledge the Reviewers and Editor for their useful comments. We have modified the manuscript to answer the questions raised by the Reviewers. Our answers, some additional comments, and explanations are given in the text below and the changes are made in the resubmitted manuscript.

We thank you for the opportunity to improve our paper, including your comments and suggestions.

Sincerely,

The authors

Answers to the Reviewer’s Comments

Answers to the First Reviewer

This paper designed a SSPP based biosensor for oil detection. It is very interesting for microwave, millimeter-wave, and terahertz bio-sensing. However, some comments should be considered for further improvement.

Our reply: We appreciate the careful reviewing of our manuscript and we gratefully acknowledge the Reviewer. Our answers and additional explanations are given bellow.

Comment 1: The title shows the bio-sensor is used for the liquid analyte detection, however in the text, only four oil samples is measured.

Our reply: Considering the Reviewer's valuable remark, the authors would like to give an additional explanation for the research design.

The proposed sensor presents proof of the concept that combines microwave sensing based on phenomena of spoof surface plasmon polaritons (SSPPs) with a microfluidic approach. In that sense, the title points out the main sensor properties and novelty. The general term ‘liquid analyte’ was used in the title since the application of the sensor can be additionally widened and optimised for other samples i.e. other ranges of dielectric constants. The proposed solution was optimized for oil samples and low dielectric constants, and therefore four oil samples available on the market were measured with the sensor in experiments and analysed according to their dielectric properties in simulations. In order to examine the sensitivity of the sensor, four additional samples of oil mixtures were prepared in different ratios, in order to detect small changes in the dielectric constant. In that manner, the sensor can be used for oil quality control with very good linearity and sensitivity.

Although the biosensors widely use oil samples, the authors would like to note that the proposed sensor is not a biosensor considering that it does not contain a biologically sensitive element, nor does the title indicate that the biosensor is proposed in the manuscript.

Comment 2: Section 2, Fig.1, it is a principle prototype. I want to know how to integrate it into the final devices or equipment?

Our reply: The authors are grateful for the valuable Reviewer's remark since the sensor integration into more complex systems is one of the possible directions for further research.

In this paper, we proposed a sensor prototype realized in simple and rapid fabrication technology based on the combination of laser micromachining and xurography. In this hybrid approach, complex designs and sensor prototypes can be realized in a cost-effective and simple way, and analysis of theory and experiments can be examined in detail without time-consuming manufacturing procedures and protocols. However, the proposed solution can be also realized in other technologies, as a multilayer printed circuit board (PCB), Low-Temperature Co-fired Ceramics (LTCC), or others, and in that way, it can be integrated together with electronic components. This approach opens the possibility for the development of multifunctional platforms for monitoring different parameters and samples important for a wide range of applications in quality control, healthcare, etc.

Comment 3: Fig.2, for the proposed biosensor structure, it seems the advantage of compactness is not very obvious, which is given in line 95 page 2. For the proposed structure, such as d=2.5mm, its bandwidth is much wider than the comb one, but from Fig.2(f), the authors just use a narrow band (3-5.6GHz). So I don't know the proposed structure is compact or not.

Our reply: We thank the Reviewer for the valuable comment.

We would like to note that the compactness lay in the fact that the proposed and comb unit cell have the same footprint area, however, the proposed cell exhibits lower SPP frequency, and this is seen in Figure 2. Due to the lower SPP frequency, the proposed configuration exhibits lower transmission zero and consequently has a narrower bandwidth in comparison to that based on the comb unit cell. Therefore, the proposed unit cell indeed exhibits compactness.

The narrow band of 3 - 6.5 GHz is used since our sensing method is based on the shift of the transmission zero, which is located in the named band for the dielectric constants that are measured.

Comment 4: Page 6, the sensor is measured with oil samples. The dielectric constant is only 1.8-2.0. So I want to know how is the performance if the liquid is changed to some other samples such as water?

Our reply: The authors are grateful for the Reviewer’s comment. An additional explanation is given below, and a new paragraph is included in the revised manuscript.

In general, microwave sensors suffer from low sensitivity and an inability to detect small changes in the dielectric constant. In this paper, the SSPP based sensor was proposed with high sensitivity properties that enabled real-time detection of small changes in the analysed sample, and that is one of the advantages of the proposed sensor.

Indeed, the proposed configuration with optimized parameters is realized for the narrow range of dielectric constants suitable for application with oils, and for that range, the sensor shows good sensitivity and linearity. Nevertheless, the sensor can be adjusted to different and wider ranges of dielectric constants. To that end, the configuration should be modified and finely tuned in terms of unit cell dimensions and substrate parameters, which, however, does not present an extensive and complicated procedure.

Line 183-188

It should be noted that the presented sensor dimensions have been optimized for excellent sensing properties for the narrow range of dielectric constants of different oil samples. Nevertheless, the sensor can be adjusted to different and wider ranges of dielectric constants. To that end, the configuration should be modified and finely tuned in terms of unit cell dimensions and substrate parameters, which also opens a possibility for further sensor miniaturization.

Comment 5: Table 1, the dielectric constant and loss is estimated. Please give the data support.

Our reply: The authors appreciate the Reviewer's suggestion.

The values of the dielectric constants and losses in Table 1 for palm, sunflower, and castor oils are given based on the data found in literature, which is well documented in the references.

In general, in the GHz range, the dispersion of dielectric constant values of olive and sunflower oil have similar behavior [69] and the oil quality determines the values of the dielectric constant. For that reason, we could expect a small difference in the sensor response for sunflower and olive oils, which may contain different responses in comparison to literature values, due to the oil quality differences. In order to examine the sensor capability to detect a small difference in dielectric constant detection in the experiments, four additional samples of oil mixtures were prepared. The parameters for these samples were calculated according to the Kraszewski formula [66], which predicts the dielectric properties of two-phase mixtures based on their ratio in the oil mixture, and they are presented in Table 1.

Comment 6: Fig.6, the simulation and measured results agree not well. Please explain the reasons.

Our reply: The authors are grateful for the comment An additional explanation is given below and involved in the revised manuscript.

The small difference between simulations and measured results can be a consequence of losses that exist in connectors and cables, and the imperfection of fabrication technology. Namely, the laser beam width causes a small difference between the modelled and fabricated structure, and that can also lead to a small spectral shift.

Line 255-258

Figures 5 and 6 present simulated and measured responses for pure edible oil samples and mixed oil samples, respectively. One should note that the simulated responses have been obtained using the calculated dielectric constants in Table 1. It can be seen that spectral positions of transmission zeros are shifted towards lower frequencies for increasing values of samples’ dielectric constants. Also, a good agreement between simulation and measurement results can be observed, except for a small spectral shift and higher losses, where the latter can be attributed to the losses in cables and connectors and imperfection of the fabrication procedure. Since the spectral shift is small, it does not influence the sensor performances to a great extent.

Comment 7: Fig.7, the results are fitted from the transmission zeros with samples of oil. If the samples is changed to some other liquid (its dielectric constant may change) or the oil mixture is in-homogeneous, the transmission may disappear. So the bio-sensor will not work well.

Our reply: The authors thank the Reviewer for this comment. In the paragraph below, we discuss the Reviewer's conclusion.

The proposed SSPP based sensor measures the transmission coefficient for the sample inside the chamber in real-time. As it was previously explained, the proposed configuration was optimised for the values of dielectric constant that correspond to the oil samples, and it can be further modified for other ranges of dielectric constant. In other words, the sensor configuration can be adjusted for the expected measurement range. Thus, the transmission zero can disappear only if the dielectric constant of the sample is not in the specified measurement range.

Comment 8: Ref.[29], the format needs to be improved.

Our reply: The authors are grateful for the detailed review and this observation in the paper. The reference [29] was corrected in the revised manuscript.

Reviewer 2 Report

Dear Authors,

Thank You for the opportunity of reading this article.

General statements about the article:

-> The article discusses a microwave microfluidic sensor based on spoof surface plasmon polaritons for ultrasensitive detection of dielectric constant. Thus the topic and scope of the article are interesting, actual, and highly desirable.

-> The article content suite to Sensors journal scope.

-> abstract is adequate to article content

-> Keywords are correctly proposed.

-> Literature review is based on 69. They are related to article content. A sufficient number of them are actual.

-> quality of the presentation is fine.

However, I indicated the following elements to revision:

#1

In the introduction, a lot of references are pointed – 65 positions. But in fact, there is no deep analysis of them. So I suggest discussing at least 15 (main problems, main results, etc.) of them to highlight the contribution of this paper in point of the novelty.

#2

Please extend conclusions. I suggest to add a paragraph with limitations of the proposed approach as well as future research directions in conclusions section.

#3

Please also revise the manuscript regarding the personal way of addressing the text. Please avoid and replace we" or "our" with the impersonal manner of addressing. The text will sound much more professional.

#4

Please extend the introduction with the description of the article. I mean “section 2 presents…, section 3 concerns… etc.”

#5

The structure of the article - The section "Theoretical and numerical analysis" should be the first part (subsection) of the "Materials and methods" section.

#6
Figure 2. should be divided into several figures - a comprehensive description of each part (from Fig. 2a to Fig. 2f) is included in the text.

Best regards,

Reviewer

Author Response

Dear Sir/Madam,

We appreciate the careful reviewing of our manuscript and we gratefully acknowledge the Reviewers and Editor for their useful comments. We have modified the manuscript to answer the questions raised by the Reviewers. Our answers, some additional comments, and explanations are given in the text below and the changes are made in the resubmitted manuscript.

We thank you for the opportunity to improve our paper, including your comments and suggestions.

Sincerely,

The authors

Answers to the Reviewer’s Comments

Answers to the Second Reviewer

Dear Authors,

Thank You for the opportunity of reading this article.

General statements about the article:

  • The article discusses a microwave microfluidic sensor based on spoof surface plasmon polaritons for ultrasensitive detection of dielectric constant. Thus the topic and scope of the article are interesting, actual, and highly desirable.
  • The article content suite to Sensors journal scope.
  • Abstract is adequate to article content
  • Keywords are correctly proposed.
  • Literature review is based on 69. They are related to article content. A sufficient number of them are actual.
  • Quality of the presentation is fine.

Our reply: The authors acknowledge the Reviewer's detailed review of the paper. Responses and additional explanations are given in below.

However, I indicated the following elements to revision:

Comment 1: In the introduction, a lot of references are pointed – 65 positions. But in fact, there is no deep analysis of them. So I suggest discussing at least 15 (main problems, main results, etc.) of them to highlight the contribution of this paper in point of the novelty.

Our reply: We thank the Reviewer for the valuable comment.

Indeed, the manuscript has a number of references and an appropriate discussion has to be provided. We would like to note that a detailed review of state-of-the-art sensors based on SPP and SSPP was presented in the introduction, as well as their applications. The properties and novelty of the proposed sensor were described in detail in Section 4 (Results and Discussion), where the sensor performances were compared with other relevant parameters of the sensors with similar applications from the literature, Table 2. In Section 4, the advantages and disadvantages of the proposed technology and sensing solution were described, and the point of novelty was highlighted.

However, according to the Reviewer's remark, the authors provide an additional paragraph in the introduction with the description of the realized sensors based on SSPP. This paragraph will introduce readers in the topic of rarely proposed SSPP based sensors with their ways of realization, advantages, and potential applications.

Line 44-60

Different planar designs that support SSPP propagation were proposed in the literature based on gradient holes [46], corrugations [47,48], zigzag grooves [48], grooves with circular patches [49], and disk resonators [50,51]. In addition, SSPP properties were used for the realization of filters [52–54], antennas [55,56] and waveguide systems [57–59]. In spite of their high potential for sensing applications, only a few sensors based on SSPP were proposed in the literature including refractive index and thickness sensor [60], a sensor for human skin tissue water content [61], and the detector of Schottky diode [62]. All of the proposed SSPP based sensing solutions [60-62] use comb structure unit cells with high sensitive properties. The sensor for human skin tissue water content [61] based on planar plasmonic waveguide was realized in printed circuit board (PCB) technology and showed the ability for in-vivo measurements of the water content in the skin tissue with potential for early diagnostic applications. On the other hand, the Schottky diode detector based on SSPP was also realized in PCB technology where the SSPP configuration enabled a significant increase in detection sensitivity [62]. Although the sensing solution based on metamaterial corrugated metal stripe structure showed ultrasensitive properties for thickness and refractive index sensing, the sensor was not realized experimentally [60]. In addition, SPP-like phenomena was proposed for dielectric constant sensing and the sensor with high sensitive performances was realized in printed circuit board (PCB) technology [9].

Comment 2: Please extend conclusions. I suggest to add a paragraph with limitations of the proposed approach as well as future research directions in conclusions section.

Our reply: The authors are grateful for the Reviewer’s suggestion. An additional paragraph with the limitations of the proposed approach, and future research directions is added in the revised manuscript.

Line 183-188

It should be noted that the presented sensor dimensions have been optimized for excellent sensing properties for the narrow range of dielectric constants of different oil samples. Nevertheless, the sensor can be adjusted to different and wider ranges of dielectric constants. To that end, the configuration should be modified and finely tuned in terms of unit cell dimensions and substrate parameters, which also opens a possibility for further sensor miniaturization.

Line 302-309

In this paper, a very sensitive and low-cost SSPP-based sensor was proposed for the detection of small changes in the dielectric constant of liquids. It is based on a novel compact unit cell, whose behaviour and sensing potential were analysed in details. The unit cell is a core element of the SSPP microwave configuration which was combined with a microfluidic channel in a multilayer structure, providing a sensing platform for liquids. The sensor was realized in the hybrid fabrication technology that combines xurography and laser micromachining processes and lamination of the realized layers. The proposed technology enables the rapid and simple realization of complex structures and provides reliable and reproducible structures. However, like in every top-down fabrication approach, the size limitations of the technology are determined with the tool used for cutting, i.e. the laser beam width, which limits the size of the realized structures. In addition, further sensor integration into complex systems and devices may require the change of fabrication technology.

To validate the sensing potential in a real-world problem, dielectric constants of edible oil samples were detected and compared to those from the literature. It was shown that the sensor exhibits excellent performance outperforming other solutions in terms of fabrication complexity, cost, and sensitivity. In addition, it is shown that sensing based on the SSPP principle opens a possibility for very sensitive applications, and by changing unit cell dimensions or substrate, the proposed sensor can be also adapted for precise measurements of gas or liquid mixtures, control of petroleum product, and food quality control. Therefore, future research will be focused on the novel sensing applications of the SSPP phenomena, as well as the sensor integration into multifunctional systems.

Comment 3: Please also revise the manuscript regarding the personal way of addressing the text. Please avoid and replace we" or "our" with the impersonal manner of addressing. The text will sound much more professional.

Our reply: The authors are grateful for the suggestion. The text was changed in the revised manuscript.

Comment 4: Please extend the introduction with the description of the article. I mean “section 2 presents…, section 3 concerns… etc.”

Our reply: The authors agree with the Reviewer’s suggestion. An additional paragraph is added to the introduction.

Line 82-89

The paper is organised into 5 sections. After the introduction with the detailed literature overview and the topic description, section 2 describes the sensor operating principle, its modelling, and simulations. Realization of the sensor through materials and method description was presented in section 3. The experimental results and comparison with simulations are presented in section 4, as well as discussion and comparison of the proposed sensor with other sensing solutions proposed for similar applications. Finally, section 5 presents a summarized conclusion for this research and future research directions.

Comment 5: The structure of the article - The section "Theoretical and numerical analysis" should be the first part (subsection) of the "Materials and methods" section.

Our reply: The authors are grateful for the Reviewer's suggestion. However, we would like to note that, due to the topic, the proposed organisation is more systematic and easier to follow for the readers.

Comment 6: Figure 2. should be divided into several figures - a comprehensive description of each part (from Fig. 2a to Fig. 2f) is included in the text.

Our reply: We thank the Reviewer for the suggestion. However, the authors believe that having all the graphs in one Figure enables comparison and analysis easier for the readers.

Reviewer 3 Report

The article is devoted to the use of spoof surface plasmon polaritons for measuring the dielectric constant of liquids at microwave frequencies. The proposed configuration of the unit cell makes it possible to reduce the dimensions of the sensor in comparison with the classical comb-cell while maintaining the value of the analysis frequency.

  1. Lines 123-124: «….which is due to the higher effective inductance and consequently lower resonant frequency». It can also be explained by a decrease in the Bragg frequency with an increase in the period d of the considered periodic structures, which in fact are SSPP.
  2. Lines 160-161: «Namely, at the surface plasmon frequency propagation constant tends to infinity and thus the propagation is not allowed». Rather, the group velocity tends to zero, so there is no wave propagation.
  3. Why is the number of cells equal to 7, how does their number affect the resonance frequency and measurement sensitivity? It will be good to show the influence of number of cells on resonance frequency.
  4. Does the proposed method allow measuring dielectric losses? The article deals only with the real part of the complex dielectric constant.

Round 2

Reviewer 2 Report

Dear Authors,

Thank you for preparing the revised version of the article. All my previous comments are included or discussed. I recommend to publish the paper in the present form.

Best regards,

Reviewer

Reviewer 3 Report

I thank the authors for the detailed response. I agree with all answers.